# A New Trial to Measure ABO Antibodies Using Complement-Dependent Cytotoxicity

**DOI:** 10.3390/medicina58060830

**Published:** 2022-06-20

**Authors:** Hee-Jeong Youk, Ho-yoon Ryu, Suk Won Seo, Jin Seok Kim, Yousun Chung, Hyungsuk Kim, Sang-Hyun Hwang, Heung-Bum Oh, Won-Ki Min, Dae-Hyun Ko

**Affiliations:** 1Department of Laboratory Medicine, Asan Medical Center, University of Ulsan College of Medicine, Seoul 05505, Korea; jiyeonhwastar@gmail.com (H.-J.Y.); fbghdbss@naver.com (H.-y.R.); seosk@amc.seoul.kr (S.W.S.); amcbloodbank@naver.com (J.S.K.); mindcatch@hanmail.net (S.-H.H.); hboh@amc.seoul.kr (H.-B.O.); wkmin@amc.seoul.kr (W.-K.M.); 2Department of Laboratory Medicine, Kangdong Sacred Heart Hospital, Seoul 05355, Korea; yousun623@gmail.com; 3Department of Laboratory Medicine, Seoul National University Hospital, Seoul 03080, Korea; hyungsuk.kim79@gmail.com

**Keywords:** ABO antibody, transplantation, complement-dependent cytotoxicity, titration

## Abstract

*Background and objectives*: The ABO antibody (Ab) titration tests are used in monitoring in ABO-incompatible (ABOi) solid organ transplantation (SOT). However, currently developed ABO Ab tests show Ab binding reactions. This study attempted to measure ABO Ab level using complement-dependent cytotoxicity (CDC). *Materials and methods*: We studied 93 blood group O serum samples from patients who underwent ABOi SOT from January 2019 to May 2021. Patients’ sera were incubated with A1 or B cells and added to a human complement solution. Supernatants were collected after centrifugation, and free hemoglobin (Hb) was measured by spectrophotometry. We converted plasma Hb value to hemolysis (%), which were compared with ABO Ab titer. *Results*: We found a mild correlation between hemolysis and ABO Ab titers. In simple regression analysis, the correlation coefficients were within 0.3660–0.4968 (*p* < 0.0001) before transplantation. In multiple linear regression analysis, anti-A hemolysis (%) was higher in immunoglobulin M (IgM) (β = 12.9) than in immunoglobulin G (IgG) (β = −3.4) (R2 = 0.5216). Anti-B hemolysis was higher in IgM (β = 8.7) than in IgG (β = 0.0) (R2 = 0.5114). There was a large variation in hemolysis within the same Ab titer. *Conclusions*: CDC can be used in a new trial for ABO Ab measurement. Furthermore, IgM rather than IgG seems to play a significant role in vivo activity, consistent with previous knowledge. Thus, this study may help in the development of the ABO Ab titration supplement test for post-transplant treatment policy establishment and pre-transplant desensitization.

## 1. Introduction

ABO antibodies (Abs) are naturally occurring and mainly immunoglobulin M (IgM) Abs, except in group O individuals whose Abs are mostly immunoglobulin G (IgG) [1]. Among IgG subclasses, IgG2 is considered the main component of ABO Abs, like other Abs against carbohydrate antigens [2,3,4]. ABO Abs are clinically responsible for potential risks of hemolytic transfusion reaction, ABO-incompatible (ABOi) hemolytic disease of fetus and newborn, delayed engraftment in ABOi hematopoietic stem cell transplantation, and rejection in ABOi solid organ transplantation (SOT) [5,6].

Measurement of ABO Ab level is essential for successful ABOi SOT. The ABO Ab titration test evaluates the effectiveness of desensitization with ABOi SOT and is an essential test in deciding the period to perform transplantation. Additionally, the titer value helps in the diagnosis of Ab-mediated rejection after ABOi SOT and determines whether to perform additional desensitization [7,8]. Although flow cytometry or enzyme-linked immunosorbent assay have been developed, they are not widely used and cannot replace ABO Ab titration methods [9].

ABO Ab titration test has some inherent flaws. It has not been standardized yet, and there are significant interobserver and interlaboratory variations [5,10]. Titration techniques used presently are different among laboratories—e.g., tube vs. column, IgM vs. IgG [11,12]. The semiquantitative nature of the titration test is another disadvantage for clinical use. Lastly, like other currently developed techniques, it gives information about the antigen–Ab binding reaction, which is insufficient regarding events in vivo. The complement-dependent cytotoxicity (CDC) method introduced in the 1960s has been widely used as a standard test method for human leukocyte antigen (HLA) serotyping [13]. Presently, it is used for crossmatching. The CDC method has the advantage of reflecting in vivo activity. If an ABO Ab test using CDC-like HLA serotyping can be developed to reflect the actual activity in vivo, in that case, it is expected to be a better indicator than isoagglutinin titration method for establishing a treatment policy after transplantation and desensitization before transplantation.

This study aimed to investigate the following: (1) quantitative ABO Ab measurement assessment using CDC and (2) whether ABO isoagglutinin IgM or IgG is more important in estimating ABOi SOT.

## 2. Materials and Methods

This study was approved by the institutional review board of the Asan Medical Center (2018-1585). The need for consent was waived by the ethics committee because only anonymized residual samples were used.

### 2.1. Study Samples

Residual serum samples were collected from 93 blood group O serum samples from patients who underwent or were scheduled to undergo ABOi SOT at the Asan Medical Center, a major tertiary hospital in Seoul, Korea, from January 2019 to May 2021. All included patients were Korean.

### 2.2. Measurement of Isoagglutinin Titer

IgM isoagglutination titration tests were conducted using the immediate spin tube method. Test tubes were prepared, and 100 µL of 0.9% normal saline were aliquoted into each tube. Moreover, 100 µL of the patient’s serum were serially diluted twofold several times. After adding 100 µL of 3% Affirmagen A1 or B red cell suspension (Ortho Clinical Diagnostics, Raritan, NJ, USA) and mixing well, each test tube was maintained at room temperature for at least 10 min and centrifuged for 15 s at 2540× *g*. We interpreted the IgM isoagglutinin titer when the highest serum dilution tube showed agglutination >1+. Thus, the highest fold in which agglutination occurs was used as the agglutinin titer. IgG isoagglutinin titer was measured at 30 min incubation at 37 °C after addition of antiglobulin reagents. Furthermore, two drops of polyspecific anti-human globulin (Millipore, Livingston, UK) were added and mixed well. After centrifugation at 2540× *g* for 15 s, the titer endpoint that showed grade 1+ agglutination was interpreted as the IgG isoagglutinin titer.

### 2.3. Plasma Hb Level Measurement Using CDC Test and Hemolysis (%) Calculation

We prepared washed reagent cells using 3% Affirmagen A1 or B cells (Ortho Clinical Diagnostics) and phosphate-buffered solution (PBS, Dulbecco’s phosphate-buffered saline MgCl_2_ and CaCl_2_, liquid, sterile-filtered, suitable for cell culture/D8662, Sigma-Aldrich, St. Louis, MO, USA). We put the same amount of PBS into A1/B cell in the tube and mixed it by inversion and then centrifuged it at 2540× *g* for 20 s. After discarding the supernatant, we repeated this process to wash the A1/B cells twice to remove the potential interferents in the reagent cells that might hinder complement activation. Washed cells were resuspended in the same amount of PBS.

Before RBC cell hemolysis using the complement, we prepared Gelatin HEPES Buffer (GHB) and complement stocks. The GHB was made of 5 mM HEPES (4-(2-hydroxyethyl) piperazine-1-ethanesulfonic acid, N-(2-hydroxyethyl) piperazine-N′-(2-ethanesulfonic acid)/H4034-100G (Sigma-Aldrich), 71 mM NaCl (Ducksan Science, Seoul, Korea), 0.15 mM CaCl_2_ (Ducksan Science), 0.5 mM MgCl_2_ (Ducksan Science), 0.1% gelatin (Sigma-Aldrich), and DW. We made complement stocks as follows: First, we added 1 mL cold DW to the human complement serum (HUMAN COMPLEMENT SERUM S1764, Sigma-Aldrich) and inverted it to dissolve well. Second, we diluted the complement in DW 1:100 using cold GHB.

As shown in Figure 1, the serum of patients with blood type O (250 µL) was incubated with the same quantity of prepared A1/B cells (250 µL) for 30 min at 37 °C. Next, we added a complement stock solution (250 µL). After incubation at 37 °C for 2 h, supernatants were collected after 15 s of centrifugation at 2540× *g*. After mixing with 300 µL of supernatants and 3 mL of 0.01% Na_2_CO_3_ (Sigma-Aldrich), free Hb level was measured using a UV-Vis spectrophotometer U-3310 (Hitachi, Tokyo, Japan) at 380–415–450 nm. We used the Harboe equation: free Hb (mg/dL) = {(A415 nm × 2) − A380 nm − A450 nm)} × 83.6 [14,15]. For calculation of hemolysis (%), 100% positive control was made by mixing A1/B cells and PBS (1:2) with freezing and thawing repeated three times for 30 min at −80 °C. Negative control was used, and 0.9% NaCl (250 µL), 3% Affirmagen A1/B cell (250 µL), and human complement serum (250 µL) were prepared as mentioned above.

We converted plasma Hb value to hemolysis (%). The formula for calculating the plasma Hb concentration (mg/dL) is as follows:(1)$$\mathrm{Hemolysis}\;\left(\%\right)=\frac{\left(\mathrm{Sample}\;\mathrm{Free}\;\mathrm{Hb}-\mathrm{Negative}\;\mathrm{Control}\;\mathrm{Hb}\right)}{\left(\mathrm{Positive}\;\mathrm{Control}\;\mathrm{Hb}\;-\mathrm{Negative}\;\mathrm{Control}\;\mathrm{Hb}\right)}\times 100\left(\%\right)$$

These values were compared with ABO Ab titer measured as described above, as anti-A IgM, anti-A IgG, anti-B IgM, and anti-B IgG.

### 2.4. Statistical Analysis

Statistical analysis was conducted using MedCalc software version 9.0 (MedCalc Software, Mariakerke, Belgium) and R version 4.1.0 (R Project for Statistical Computing, Vienna, Austria). The correlations between anti-A IgM and anti-A IgG titer and anti-B IgM and IgG titer were analyzed. Multiple comparisons were conducted for each ABO Ab titer to hemolysis values (%). Multiple forward stepwise linear regression analysis was conducted to determine the relationships among variables. A *p*-value < 0.05 was considered statistically significant.

## 3. Results

Among 93 blood group O serum samples, 72 were collected during the desensitization period before transplantation (50 for anti-A IgM, 44 for anti-A IgG, 48 for anti-B IgM, and 43 for anti-B IgG), and 21 were acquired after transplantation (19 for anti-A IgM, 7 for anti-A IgG, 12 for anti-B IgM, and 6 for anti-B IgG). Groups A and B samples were excluded because most ABO Abs in the groups are IgM and the number of samples was small.

### 3.1. Hemolysis (%) According to ABO Ab Titers

The comparison of hemolysis (%) and ABO Ab titers, such as anti-A IgM, anti-A IgG, anti-B IgM, and anti-B IgG, is summarized in Figure 2.

The data reveal a tendency that hemolysis (%) increases as titer increases, for both IgM and IgG. Samples with low titer ABO Ab, such as lower than 4 for IgM or lower than 32 for IgG, showed almost no hemolysis at all. Although we observed a mild correlation between hemolysis (%) and ABO Ab titers, there was a large variation in hemolysis (%) within the same Ab titer. For example, samples with anti-A IgM titer of 1:128 showed hemolysis (%) from near 0% to almost 100% (Figure 2A).

Simple linear regression analysis is shown in Table 1. Before transplantation, anti-A IgM, anti-A IgG, anti-B IgM, and anti-B IgG were significantly associated with hemolysis (%) (*p* < 0.0001). However, after transplantation, anti-A IgM, anti-A IgG, anti-B IgM, and anti-B IgG were not significantly associated with hemolysis (%) (*p*, 0.0583–0.2740).

### 3.2. Relative Contribution of IgM and IgG for Hemolysis Activity

Figure 3 shows 3D scatter plots that display the correlation between hemolysis (%), IgM titer, and IgG titer. Hemolysis (%) was correlated with both IgM and IgG titers, as expected. As it is difficult to find which of the titers is more important for determination of hemolysis (%) in 3D scatter plots, we conducted multiple regression analysis. In the multiple regression analysis (Table 2), anti-A hemolysis (%) was more correlated with IgM (β = 12.9, *p* = 0.0018) than with IgG (β = −3.4, *p* = 0.3752) (R^2^ = 0.5216). Anti-B hemolysis (%) was also more correlated with IgM (β = 8.7, *p* = 0.0110) than with IgG (β = 0.0, *p* = 0.9889) (R^2^ = 0.5114). Thus, hemolysis values (%) of IgM titers were more significant than hemolysis values (%) of IgG titers. However, the overall R2 values are somewhat low (0.5226 and 0.5114). These low values imply that the complement activation capacity is also affected by unknown factors other than isoagglutinin titer values.

### 3.3. Comparison between Pre- and Post-Transplantation Samples

The range and median of hemolysis values (%) corresponding to the matched ABO Ab titer between pre-transplantation and post-transplantation were noted (Table 3). Generally, the hemolysis values (%) are higher before transplantation than those after transplantation. We did not perform statistical analysis because the sample size in each group is small. Although the overall titer or hemolysis values (%) after transplantation is low, ABO Ab titers of some specimens showed high activity (up to 23.4%). However, there were no clinical rejection episodes in these patients. Moreover, some samples with a high titer, for example, IgG anti-B titer of 1024, showed minimal hemolysis (2.85%) after transplantation.

## 4. Discussion

We found mild correlations between hemolysis (%) and ABO Ab titers. IgM titers showed higher hemolysis values (%) than IgG titers. About two weeks after transplantation, the accommodation from even high anti-A/B Ab exposure does not harm the organ transplant [16,17]. Although the overall titer or hemolysis values (%) after transplantation is low, ABO Ab titers of some specimens showed high activity. However, no clinical rejection was noted in patients after transplantation in this study.

ABO Ab titration methods have been used for a long time in monitoring patients in ABO-incompatible SOT. The limitations of the currently used ABO Ab titration method are as follows: (1) difference in methods between laboratories, (2) unstandardized method, and (3) semiquantitative ABO Ab titration [6,12,18,19,20,21]. Our CDC method answers some of these questions. It can be more easily standardized and generates quantitative numeric values. These characteristics have virtues in clinical practice over titration methods. For example, some patients show one titer elevation after transplantation, but we cannot confirm whether this indicates true elevations in Ab levels. Due to the nature of titrations methods, we can assure the elevation of Ab levels when there are two titer elevations.

Agglutinin titration in blood group O can be interpreted as reactions by mainly IgM and IgG [16]. It is known that IgM is the main factor in ABO Abs, while IgG2 is the main factor in IgG subclasses. However, which factor is more important is still controversial. Both IgM and IgG can activate complement. IgG1 and IgG3 can trigger the classical complement pathway with effects, but IgG2 and IgG4 do not [2,3,4]. Therefore, not IgG anti-ABO Abs but IgM anti-ABO Abs may influence Ab-mediated rejection importantly [16]. The result of this study showed that IgM was a more critical factor than IgG in ABOi SOT. Furthermore, IgG is not the main risk but a zero risk, although the sample size was small. In terms of complement activation activity, IgM titer tends to be higher than IgG titer. When IgM and IgG were used clinically, IgM has the advantage of being able to see the titer faster than IgG.

After transplantation, the ABO Ab titer itself was low, and we presume that the hemolytic activity did not increase even if the ABO Ab titer increased. It indicates that accommodation occurred. Generally, within two or three weeks after ABOi SOT, accommodation occurs, and the titer of ABO Abs increases. After transplantation, IgM and IgG remain stable, and resistance to AMR occurs, which is called accommodation [16,22]. However, there is no significant effect on the transplanted organs. Some samples showed high hemolysis values in accommodation, but most of them showed low hemolysis values. To elucidate this, we need a more extensive study.

If the CDC method is systematized and further supplemented, another application of our CDC method might be used in the side effects of hemolytic transfusions. In patients with Abs against the RBC antigen, transfusion of cognate antigen-negative blood is common, but an Ab against a high-incidence red cell antigen or a newly discovered Ab whose clinical significance is not well known is difficult to obtain antigen-negative blood. If these Abs are present, there would be difficulty in the selection of blood transfusion under the clinical situation. Although unavailable in most laboratories, the monocyte monolayer assay method can be used to predict the extravascular hemolytic response [23,24]. However, there is no adequate method to predict the intravascular hemolytic response. This CDC method is expected to be used to predict the activity of Abs to RBC antigens other than ABO Ab.

The limitations of this study were that we did not examine the distribution of IgG subclasses in the blood O group. However, we have shown that CDC can be used in a new trial for the ABO Ab measurement method. In our study, IgM seems to play a major role rather than IgG, which is consistent with the previous knowledge. Therefore, hemolytic (%) measurement of ABO Ab using CDC may help in understanding the clinical significance of the ABO Ab. Using an automated analyzer for Ab titration and free hemoglobin measurements, the ABO Ab monitoring methods would be further advanced.

## 5. Conclusions

Our study showed a new trial for ABO Ab measurement using the CDC method. In the first one, we tried to measure quantitative ABO Ab using this CDC method. In the second one, IgM seems to play a major role rather than IgG in the CDC method, which is consistent with the previous knowledge. If this CDC method is standardized with more experiments, it may be an indicator that can quantitatively measure the ABO Ab reaction. Additionally, the CDC method is expected to be useful for not only transplantation but also blood transfusion.

## Figures and Tables

**Figure 1 medicina-58-00830-f001:**
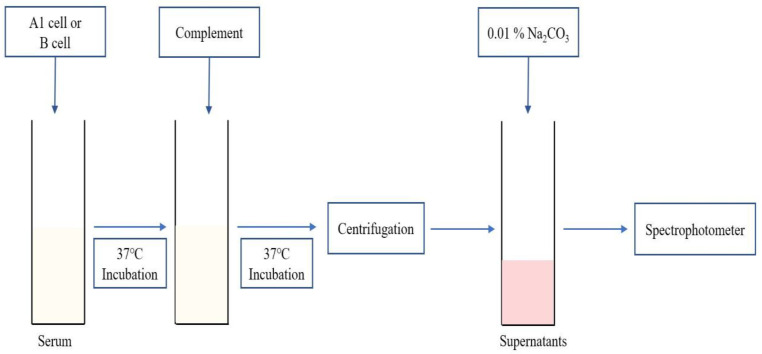
Schematic representation of the plasma Hb level measurement method.

**Figure 2 medicina-58-00830-f002:**
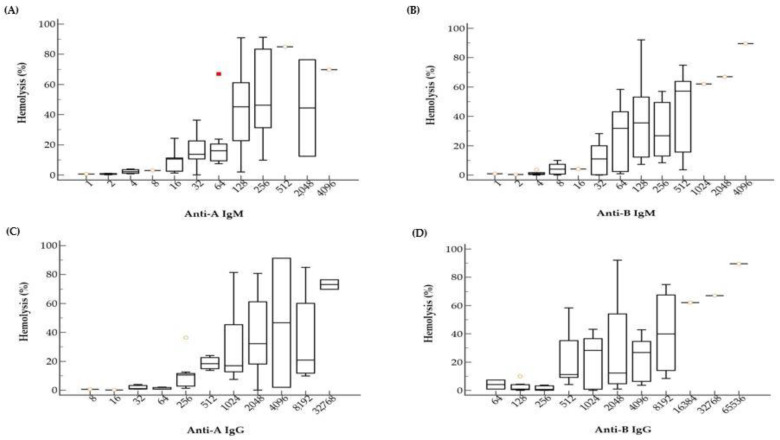
Hemolysis values (%) according to anti-A and anti-B isoagglutinin titers. As the titer increases, the hemolysis% also increased. There are substantial variances in hemolysis% among samples with the same isoagglutinin titers. Box and whiskers plots indicate hemolysis% according to anti-A IgM (**A**), anti-B IgM (**B**), anti-A IgG (**C**), and anti-B IgG (**D**) titers. The red square indicates an outlier.

**Figure 3 medicina-58-00830-f003:**
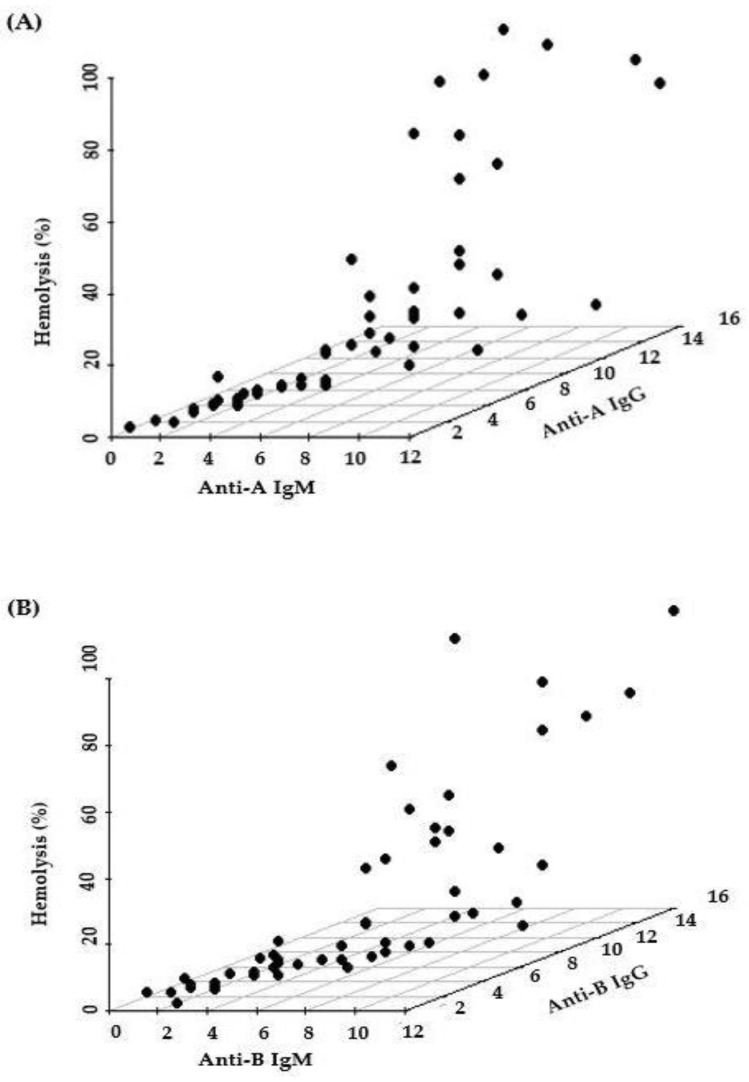
Hemolysis value (%) according to IgM and IgG of Anti-A (**A**) and Anti-B (**B**) using 3D scatterplot.

**Table 1 medicina-58-00830-t001:** Simple linear regression analysis results of hemolysis (%) related to ABO Ab titers.

Variable		Pre-Transplantation			Post-Transplantation			
	Slope	95% CI	*p*-Value	R^2^	Slope	95% CI	*p*-Value	R^2^
Anti-A IgM	23.2	15.4–31.0	<0.0001	0.4267	8.1	−3.1–16.5	0.0583	0.1952
Anti-A IgG	19.9	11.7–28.0	<0.0001	0.3660	−4.9	−15.3–5.4	0.2740	0.2318
Anti-B IgM	23.2	16.2–30.1	<0.0001	0.4968	1.2	−0.2–2.6	0.0850	0.2676
Anti-B IgG	23.3	15.0–31.5	<0.0001	0.4439	1.2	-0.4–2.8	0.0970	0.5382

CI: confidence interval; R^2^: coefficient of determination. The coefficient of determination (R^2^) is the degree of variation explained by the simple linear regression model based on the total variation of the dependent variable.

**Table 2 medicina-58-00830-t002:** Multiple linear regression analysis results of hemolysis (%) related to ABO Ab titers.

Variable	Multiple Linear Regression			
	Coefficient	t	*p*-Value	R^2^
Anti-A IgM	12.9	3.282	0.0018	0.5216
Anti-A IgG	−3.4	−0.894	0.3752	
Anti-B IgM	8.7	2.631	0.0110	0.5114
Anti-B IgG	0.0	−0.014	0.9889	

R^2^: coefficient of determination.

**Table 3 medicina-58-00830-t003:** Hemolysis range (%) according to ABO Ab titer before and after transplantation. The titer indicates ABO Ab titer of agglutinin titration test in blood group O.

Variable	Titer	Pre-Transplantation		Post-Transplantation	
		Hemolysis Range (%)	Median (%)	Hemolysis Range (%)	Median (%)
Anti-A IgM	2	0.11–1.11	0.73	0.00–2.79	0.71
	4	0.80–3.93	2.14	0.10–13.02	1.77
	8	2.94	2.94	1.11–23.40	3.28
	16	1.30–24.31	10.58	7.25–9.16	7.25
Anti-B IgM	1	0.89	0.89	0.20–1.28	0.74
	2	0.391	0.39	0.15–1.52	0.87
	4	0.00–3.72	0.83	0.00–2.34	1.42
	8	0.00–10.01	4.01	1.20–1.26	1.23
	16	4.13	4.13	0.00–2.20	1.10
	32	0.00–28.25	11.00	2.85	2.85
Anti-A IgG	8	0.61	0.61	2.79	2.79
	32	0.73–3.93	1.11	1.30–10.31	5.41
	64	0.74–2.14	0.80	0.00–1.89	0.81
	256	1.30–36.34	10.51	1.11	1.11
Anti-B IgG	128	0.00–10.01	0.89	0.00	0.00
	256	0.00–3.72	0.68	2.20	2.20
	512	4.13–58.38	11.16	0.00	0.00
	1,024	0.00–43.20	28.25	2.85	2.85

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
