# Peer review of "A New Trial to Measure ABO Antibodies Using Complement-Dependent Cytotoxicity"

_medicina, 2022, doi:10.3390/medicina58060830_

Round 1

Reviewer 1 Report

In this paper, Hee-Jeong Youk and coll. describe the correlation between ABO antibody titer detected by the classical method and a direct measure of hemolysis by the CDC test.

The authors have clearly described the lack of a reliable and standardized test for the measurement of iso-hemagglutinin and the test proposed in this research could be a valid help.

The results reported in this research are interesting however they need validation in a greater number of patients of all blood groups; it is also mandatory to expand the sample of transplanted patients.

English could be revised

Author Response

Response to Reviewer #1

Thank you for reviewing our manuscript. We have answered each point below.

Comment: In this paper, Hee-Jeong Youk and coll. describe the correlation between ABO antibody titer detected by the classical method and a direct measure of hemolysis by the CDC test.

The authors have clearly described the lack of a reliable and standardized test for the measurement of iso-hemagglutinin and the test proposed in this research could be a valid help.

Major comments:

  1. The results reported in this research are interesting however they need validation in a greater number of patients of all blood groups; it is also mandatory to expand the sample of transplanted patients.

Response: Thank you for the comments. We strongly agree with your opinion that this study needs extensive validation in a larger number of patients. This is a small-scale preliminary study for the first trial of the newly developed CDC method. We would like to expand the study in the future.

  1. English could be revised

Response: The revised manuscript was checked by a professional English editing service (Editage).

Reviewer 2 Report

The study was aimed to test if complement-dependent cytotoxicity (CDC) can be used as a new trial for the ABO Ab measurement method. Specifically, the authors investigated the quantitative ABO Ab measurement assessment using CDC and whether the ABO isoagglutinin IgM or IgG is more important in estimating ABOi SOT.

The CDC method is a reliable and established method for human leukocyte antigen (HLA) serotyping and is well described in the special literature, but it is a difficult-to-operate. In fact, CDC method requires much manual skill and non-rapid execution times.

The measure ABO antibodies using complement-dependent cytotoxicity, for monitoring in ABO-incompatible solid organ transplantation is a new idea.

My comments are:

Line 69 “Materials and Methods” section: A graphical representation of this section, could improve the understanding of the whole study design. Also, the number of recruited patients of each study group should be inserted in the “Results” section.

Line 133 “Results” section:

- The description of study population characteristics is missed (age, gender, race)

- The results should be explained in more detail in the text and not only showed in the figures or tables.

- Every statistically significance should be reported and a concise explanation/caption of each figures/table should be provided.

.

Line 176 “Discussion” section:  

I would suggest to the author providing a premise to underline the importance of their trial and the impact of the results could have in the clinical practice.

The authors well described the limitations of their study, but did not refer the possibility to overcome them through automated assay for ABO antibodies tritation (e.g. Immucor) and for free haemoglobin (the use of spectrophotometer is obsolete).

Author Response

Response to Reviewer #2

Thank you for reviewing our manuscript. We reviewed, corrected, and added content based on your comments.

Comment: The study was aimed to test if complement-dependent cytotoxicity (CDC) can be used as a new trial for the ABO Ab measurement method. Specifically, the authors investigated the quantitative ABO Ab measurement assessment using CDC and whether the ABO isoagglutinin IgM or IgG is more important in estimating ABOi SOT.

The CDC method is a reliable and established method for human leukocyte antigen (HLA) serotyping and is well described in the special literature, but it is a difficult-to-operate. In fact, the CDC method requires much manual skill and non-rapid execution times.

The measure ABO antibodies using complement-dependent cytotoxicity, for monitoring in ABO-incompatible solid organ transplantation is a new idea.

Major comments:

  1. Line 69 “Materials and Methods” section: A graphical representation of this section, could improve the understanding of the whole study design.

Response: We have added Figure 1 for better understanding of the whole study design.

Also, the number of recruited patients of each study group should be inserted in the “Results” section.

Response: We inserted the number of recruited patients of each study group in the “Results” section.

“Among 93 blood group O serum samples, 72 were collected during the desensitiza-tion period before transplantation (50 for anti-A IgM, 44 for anti-A IgG, 48 for anti-B IgM, and 43 for anti-B IgG), and 21 were acquired after transplantation (19 for anti-A IgM, 7 for anti-A IgG, 12 for anti-B IgM, and 6 for anti-B IgG). Groups A and B samples were excluded because most ABO Abs in the groups are IgM and the number of samples was small.”

  1. Line 133 “Results” section:

- The description of study population characteristics is missed (age, gender, race)

Response: Because we collected residual serum samples from blood group O serum samples from patients who underwent or scheduled to undergo ABOi SOT, we did not review sex and age in the patient information. However, all patients were Korean. We have added the following sentence:

 “All included patients were Korean.”

- The results should be explained in more detail in the text and not only showed in the figures or tables.

Response: Thank you for the suggestions. We have added more explanations in the Results section.

- Every statistically significance should be reported and a concise explanation/caption of each figures/table should be provided.

Response: Thank you for the comments, and we have revised the manuscript as recommended. Statistically significant values are presented only in the multiple regression section.

“In the (Table 2), anti-A hemolysis (%) was more correlated with IgM (β = 12.9, P = 0.0018) than with IgG (β = −3.4, P = 0.3752) (R2 = 0.5216). Anti-B hemolysis (%) was also more correlated with IgM (β = 8.7, P = 0.0110) than with IgG (β = 0.0, P = 0.9889) (R2 = 0.5114).”

  1. Line 176 “Discussion” section:

I would suggest to the author providing a premise to underline the importance of their trial and the impact of the results could have in the clinical practice.

The authors well described the limitations of their study, but did not refer the possibility to overcome them through automated assay for ABO antibodies tritation (e.g. Immucor) and for free haemoglobin (the use of spectrophotometer is obsolete).

Response: Thank you for the suggestions. We have added the following descriptions in the Discussion section.

“ABO Ab titration methods have been used for a long time in monitoring patients in ABO-incompatible SOT. The limitations of the currently used ABO Ab titration method are as follows: (1) difference in methods between laboratories, (2) unstandardized method, and (3) semiquantitative ABO Ab titration [6,12,18-21]. Our CDC method answers some of these questions. It can be more easily standardized and generates quantitative numeric values. These characteristics have virtues in clinical practice over titration methods. For example, some patients show one titer elevation after transplantation, but we cannot confirm whether this indicates true elevations in Ab levels. Due to the nature of titrations methods, we can assure the elevation of Ab levels when there are two titer elevations.”

“Using an automated analyzer for Ab titration and free hemoglobin measurements, the ABO Ab monitoring methods would be further advanced.”

Round 2

Reviewer 2 Report

the manuscript has been sufficiently improved